# Implementation of an Electronic National Early Warning System to Decrease Clinical Deterioration in Hospitalized Patients at a Tertiary Medical Center

**DOI:** 10.3390/ijerph18094550

**Published:** 2021-04-25

**Authors:** Chieh-Liang Wu, Chen-Tsung Kuo, Sou-Jen Shih, Jung-Chen Chen, Ying-Chih Lo, Hsiu-Hui Yu, Ming-De Huang, Wayne Huey-Herng Sheu, Shih-An Liu

**Affiliations:** 1Department of Critical Care Medicine, Taichung Veterans General Hospital, Taichung 40705, Taiwan; clwu@vghtc.gov.tw; 2Department of Automatic Control Engineering, Feng Chia University, Taichung 40724, Taiwan; 3Department of Industrial Engineering and Enterprise Information, Tunghai University, Taichung 40705, Taiwan; 4Computer & Communication Center, Taichung Veterans General Hospital, Taichung 40705, Taiwan; jmskuo@vghtc.gov.tw; 5Department of Biomedical Engineering, Hang-Kung University, Taichung 43302, Taiwan; 6Department of Nursing, Taichung Veterans General Hospital, Taichung 40705, Taiwan; soujen@vghtc.gov.tw (S.-J.S.); f7741@vghtc.gov.tw (H.-H.Y.); 7Center of Quality Management, Taichung Veterans General Hospital, Taichung 40705, Taiwan; roro0320@vghtc.gov.tw (J.-C.C.); ylo6@bwh.harvard.edu (Y.-C.L.); mingde@vghtc.gov.tw (M.-D.H.); 8Department of Top Hospital Administration, Taipei Veterans General Hospital, Taichung 11221, Taiwan; whhsheu@vghtc.gov.tw; 9Department of Medicine, School of Medicine, National Yang Ming Chiao Tung University, Taipei 11221, Taiwan; 10Institute of Medical Technology, College of Life Science, National Chung-Hsing University, Taichung 402204, Taiwan; 11School of Medicine, National Defense Medical Center, Taipei 114, Taiwan

**Keywords:** cardiopulmonary resuscitation, clinical deterioration, early warning score, health information system

## Abstract

The National Early Warning Score (NEWS) is an early warning system that predicts clinical deterioration. The impact of the NEWS on the outcome of healthcare remains controversial. This study was conducted to evaluate the effectiveness of implementing an electronic version of the NEWS (E-NEWS), to reduce unexpected clinical deterioration. We developed the E-NEWS as a part of the Health Information System (HIS) and Nurse Information System (NIS). All adult patients admitted to general wards were enrolled into the current study. The “adverse event” (AE) group consisted of patients who received cardiopulmonary resuscitation (CPR), were transferred to an intensive care unit (ICU) due to unexpected deterioration, or died. Patients without AE were allocated to the control group. The development of the E-NEWS was separated into a baseline (October 2018 to February 2019), implementation (March to August 2019), and intensive period (September. to December 2019). A total of 39,161 patients with 73,674 hospitalization courses were collected. The percentage of overall AEs was 6.06%. Implementation of E-NEWS was associated with a significant decrease in the percentage of AEs from 6.06% to 5.51% (*p* = 0.001). CPRs at wards were significantly reduced (0.52% to 0.34%, *p* = 0.012). The number of patients transferred to the ICU also decreased significantly (3.63% to 3.49%, *p* = 0.035). Using multivariate analysis, the intensive period was associated with reducing AEs (*p* = 0.019). In conclusion, we constructed an E-NEWS system, updating the NEWS every hour automatically. Implementing the E-NEWS was associated with a reduction in AEs, especially CPRs at wards and transfers to ICU from ordinary wards.

## 1. Introduction

It is estimated that 3–9% of patients experience clinical deterioration during their hospitalization [1]. Therefore, healthcare institutes require tools to recognize patients at risk of clinical deterioration to ensure that they can deliver appropriate care at the right moment. The clinical deterioration of patients in hospitals is typically preceded by changes in vital signs in the 24 h before their cardiac arrest [1,2]. A previous report found that the Early Warning System performs well for prediction of cardiac arrest and death within 48 h [3]. Numerous modified Early Warning Systems (EWS) exist worldwide. The abovementioned EWS typically consists of similar physiologic parameters. In 2012, the UK Royal College of Physicians published the National Early Warning Score (NEWS), which consists of parameters of respiratory rate, oxygen saturation, temperature, systolic blood pressure, pulse rate, and level of consciousness [4]. In addition, the NEWS system was updated in 2017 [5].

To prevent or mitigate the clinical deterioration of patients in ordinary wards, a well-designed response system is necessary to manage events. A few studies showed that the implementation of the EWS decreased the number of clinical deteriorations [6,7]. Prompting of the EWS may quickly initiate adequate treatment and facilitate communication among healthcare workers. As a result, it can improve the clinical outcome [8,9]. Nevertheless, a comprehensive review found that the EWS had no clear evidence of benefit in decreasing mortality, cardiac arrest, length of hospital stay, or transfers to the ICU [3,9]. Recently, Bedoya et al. concluded that the NEWS was generally ignored by frontline nursing staff and implementation of the NEWS had no appreciable impact on defined clinical outcomes [10].

The EWS is typically calculated manually. Moreover, repeated calculation of the EWS is time-consuming and may be inaccurate. A study using a once-per-day modified EWS (MEWS) showed that up to 18.2% of scores were calculated incorrectly [11]. Based on the innovative electronic medical record (EMR), continuous risk scoring curves can be utilized to monitor subtle variations in vital signs and predict clinical deterioration in ordinary wards successfully [12]. In order to decrease the workload of healthcare staff and enable immediate early detection of clinical deterioration, automated calculation of the EWS based on EMR data is urgently required. Furthermore, physicians and nurses are usually extremely busy and, therefore, a strategy to minimize false alarms and eliminate alarm fatigue is also imperative.

In the current study, we constructed an electronic NEWS (E-NEWS), which was updated hourly in our Health Information System (HIS) and Nurse Information System (NIS). The alert system of clinical deterioration was implemented in ordinary adult wards. The scores were integrated into the handover documentation during nurses’ shifts. The system also sent a text message to in-charge attending physicians twice a day if the scores were above the criteria. Here, we present the results of our E-NEWS system and its influence on healthcare outcomes.

## 2. Materials and Methods

This prospective study was conducted in Taichung Veterans General Hospital (TCVGH), a 1500-bed tertiary medical center in central Taiwan. The institutional review board of Taichung Veterans General Hospital reviewed and approved the study protocol (protocol no./IRB TCVGH No: CE19197B). Written informed consent from participants was waived. All patients’ information was anonymized and deidentified prior to analysis.

The inclusion criteria were as follows: (1) patients who were hospitalized in ordinary wards and (2) were ≥20 years old. The exclusion criteria included (1) hospitalization for less than one day (≤24 h), (2) direct admission to ICU, and (3) patients who had been intubated before admission.

### 2.1. Definition of Adverse Events and Diagnosis Groups at Discharge

The “adverse event” (AE) group consisted of patients who had received cardiopulmonary resuscitation (CPR), had been transferred to an intensive care unit due to unexpected deterioration, or died. Patients without AE were allocated to the control group. Patients with a scheduled admission to the ICU after surgery and cardiac catheterization were also assigned to the control group. All the diagnoses at discharge were collected for diagnostic grouping. We used codes of the 9th and 10th editions of the International Statistical Classification of Diseases and Related Health Problems (ICD) to classify the patients into cancer, cardiovascular disorders, neurological disorders (stroke and non-stroke), respiratory disorders, diabetes, gastroenterology, and renal disorders (Appendix A).

### 2.2. On-line Electronic NEWS System

All measurements of vital signs (temperature, respiratory rate, heart rate, and blood pressure) were uploaded in real-time to our EMR. However, the EMR data did not completely match the parameters of the NEWS system (e.g., oxygenation and consciousness). We did not have precise data on all patients’ SpO2, alertness, confusion, voice, pain, and unresponsiveness. We applied a minor modification to obtain the following parameters: regarding consciousness, we defined patients as alert if they had no Glasgow Coma Scale (GCS) data or a motor response score of 6 from nursing notes; patients receiving oxygen therapy were determined by the presence of relevant therapeutic orders (e.g., nasal cannula, mask, non-invasive positive pressure ventilator). Values for the following parameters were regarded as missing data: blood pressure >300 mmHg, pulse rate >300/min, respiratory rate >50/min, SpO2 >100%, and temperature <25 or >45 °C. If no raw data of vital signs were updated at the scheduled time point, the latest values were used for calculating the NEWS. According to the abovementioned rules, we built an ETL (Extract–Transform–Load) program that automatically calculated the hourly NEWS from our EMR system’s raw data. The training and validation procedures were developed in accordance with the TRIPOD statement [13,14]. We used 99,861 admissions data from January 2007 to December 2015 for model training. A further 33,612 admissions data was obtained from January 2016 to December 2017 for external validation. The area under curve (AUC) for training and validation model is 0.961 and 0.950, respectively.

### 2.3. Establishment of E-NEWS Dashboard on HIS and NIS

We constructed the framework of the E-NEWS dashboard (Figure 1). The framework comprised three layers: data acquisition, service, and visualization. The E-NEWS was displayed on a HIS for physicians and an NIS for nurses. Each NEWS score could be drilled down to view individual parameters of the NEWS (Appendix A). For the convenience of management, the E-NEWS could also be displayed based on the different wards or diverse subspecialties (Appendix A). We could trace the sequential hourly NEWS back to the last 72 h.

### 2.4. Implementation of E-NEWS

The E-NEWS was introduced to our clinical practice in three phases. The first phase was termed “Baseline” and covered the period from 1 October 2018 to 28 February 2019. We announced the E-NEWS system to our colleagues and listened to their feedback. We continually observed on-line E-NEWS to establish whether the results correlated with clinical status appropriately. The second phase was termed “Implementation”. We held an educational program that was conducted ward by ward and division by division from 1 March to 31 August 2019. The titles of topics in the educational program included: (1) “Clinical deterioration in wards is an important issue in patient safety”, (2) “What NEWS is” and “How to read E-NEWS”, (3) “How to respond to a NEWS score above 5 and above 7”. The third phase was termed the “Intensive period”, which lasted from 1 September to 31 December 2019. When the patients’ E-NEWS scores increased to more than 4 in the last 12 h and the last E-NEWS was ≥7, we sent a text message to the cell phones of their in-charge attending physicians at 7:00 am and 5:00 pm (Appendix A). In this intensive period, we continued the educational program and sent notifications to the patients’ attending physicians. Furthermore, we requested that nurses check E-NEWS scores during their shifts (Appendix A). If NEWS scores increased and patients were unstable without appropriate management, the in-charge nurses were required to inform the on-duty physicians in the ward.

### 2.5. Statistical Analysis

For comparisons of characteristics among the three periods, the Kruskal–Wallis test was used for continuous variables and the Chi-Square test was used for categorical variables. Multivariate analysis was used to adjust the confounding factors of the demographic variables to examine the effectiveness of the implementation of the E-NEWS. Statistical significance was defined as a *p*-value less than 0.05. All the statistical analyses were performed with the Statistical Package for the Social Sciences (IBM SPSS version 22.0; International Business Machines Corp, New York, NY, USA) and R software (Version 3.4.1). With a difference of 0.56 percent total AE rate between the baseline and intensive period, the estimated sample size to demonstrate a two-tailed significance level of 0.05 with 80 percent power would require 834 subjects in each group. However, as the sample size is large, the calculated statistical power almost reaches 100%.

## 3. Results

### 3.1. Study Population

From 1 October 2018 to 31 December 2019, there were 39,161 patients with a total of 73,674 hospitalization courses under the surveillance of the E-NEWS. Some were repeatedly admitted due to planned treatments, such as chemotherapy, exacerbation of their underlying disease, or a new problem. The demographic data of all hospitalization courses are shown in Table 1. The average age was 58 ± 17 years old and 51% of patients were male. Though we could not group all cases into the eight diagnoses at discharge, the majority of patients in the current study had cancer. Cardiovascular and neurological disorders were the second and third most common diagnosis, respectively (Table 1). More than half of the patients were cared for by the Department of Internal Medicine. We implemented the E-NEWS system in three periods (baseline, implementation, and intensive periods). As shown in Table 1, there were significant, but only small, differences among the three periods in gender, cancer, and respiratory disorder.

### 3.2. Reduction in AEs in Implementing E-NEWS

The overall AE rate among all admissions was 6.06%. The subtypes of adverse events were mortality (32.8%), CPR at wards (7.2%), and transfer to ICU (60.1%). We observed a significant decrease in the percentage of AEs from 6.06% to 5.51% (*p* = 0.001). We further explored the trends of each subtype of AEs. The percentage of mortality decreased, but this did not reach statistical significance. Interestingly, CPR at wards was reduced significantly from 0.52% during “baseline” to 0.34% during the “intensive period” (*p* = 0.012). Transfer to ICU was also decreased significantly (*p* = 0.035) (Figure 2). We noted the percentage of AEs was more obviously reduced in the intensive period but not in the implantation period, in comparison with the baseline. From 1 September to 31 December 2019, the E-NEWS alert system sent out 22 ± 7 notifications at 7:00, and 31 ± 10 at 17:00 every day.

### 3.3. AEs in Different Clinical Situations

The percentage of AEs increased with age. Furthermore, the AEs decreased significantly in patients aged 60~79 years during the three periods of E-NEWS (Table 2). According to the diagnosis groups at discharge, the percentage of AEs decreased after the implementation of the E-NEWS in patients with cardiovascular and gastroenterological disorders but increased in patients with respiratory disorders (Table 2). The percentage of AEs was higher in those receiving medical service than that in patients receiving surgical service. Implementation of the E-NEWS reduced the incidence of AEs only in those patients cared for in medical departments. Using multivariate analysis, we adjusted the confounding effects of age, gender, and diagnoses at discharge. We found the percentage of AEs was reduced significantly in the “intensive period” when compared with that at baseline (*p* = 0.019, OR 0.90, 95%CI 0.83–0.98) (Table 3).

## 4. Discussion

We applied a minor modification to the NEWS in our institute and successfully created an electronic version, known as E-NEWS. The scores were automatically calculated and updated hourly. The E-NEWS was displayed on the dashboard and could be presented based on different wards or various divisions. The Early Warning Systems of clinical deterioration were integrated into the nurses’ handover documentation and the physicians’ clinical practices. Based on an analysis of the three periods of implementation, we found the percentage of clinical deterioration decreased significantly, especially CPRs and transfers to the ICU at the intensive period, which included an alarm, text message notification, and the integration of routine handover of documentation.

A study by Churpek et al. excluded all patients with scheduled procedures for AEs transferred to the ICU [13]. We excluded patients with a scheduled admission to the ICU after surgery and invasive cardiac catheterization. The AE rate among all hospitalizations was 6.06%. This was similar to previous studies, which reported rates of 6.1% and 5.1% [15,16]. Moreover, 42.7% of enrolled patients had a cancer diagnosis. However, the percentage of AEs in cancer patients was relatively low (6.66%), which was lower than those patients with the other diagnosis at discharge (Table 2). The reason seemed to be that they were repeatedly admitted due to scheduled treatments, such as chemotherapy. Those patients with respiratory disorders decreased significantly in the intensive period. The percentage of AEs increased after introducing the E-NEWS system. A reason for this might be that the characteristics of patients with respiratory disorders had been changed. In the future, we have to monitor the situation and explore the causes from the patients, or from the physicians or nurses.

Sutherasan et al. used the early warning scores at admission to stratify the patients into low, moderate, or high risk, based on the NEWS system, but found no change in the patients’ overall outcomes. The study did not assess the NEWS continuously after admission [17]. Real-time data of the deterioration risk should be immediately visible to the entire clinical team to optimize awareness of the current situation. The chronology of escalation in clinical deterioration must be reported to the care team automatically in order to enhance patient safety. The main purpose of the EWS is to identify patients at risk of clinical deterioration, especially in the acute healthcare ward, and should therefore be simple, reliable, and comprehensible for all healthcare providers in the institute [18,19]. The framework of our E-NEWS can provide information hourly and automatically, and thus, meets the key requirement of an early warning system.

Using 16 semi-structured interviews, de Vries et al. showed that EWS facilitate communication between nurses and resident doctors. Nurses also feel more confident to contact doctors and demand action [20]. We integrated the E-NEWS into the NIS and instructed nurses to check the scores during their shifts. This design introduced a novel aspect into the routine work of our nurses. One of the head nurses gave us feedback that her staff could glean all the pertinent information related to the severity and urgency of all patients at a glance on the E-NEWS dashboard; they were more confident to discuss treatment plans with the physicians. Therefore, hourly E-NEWS could be an effective platform for communication between nurses and physicians.

A previous study indicated that attentive and skilled physicians may fall short of their best performance [1]. This could be explained by the busy schedules of doctors and the lack of reminding systems. Because our E-NEWS system provided scores hourly, the care team was able to identify the risk of clinical deterioration for hours, or even days. In addition to alerting staff to the development of acute collapse, our E-NEWS system could provide information on progressive clinical deterioration, based on the chronology of E-NEWS scores. We also sent a notification of clinical deterioration directly to in-charge physicians. They then had enough time and adequate information to apply an appropriate treatment plan, such as palliative care for cancer patients at the terminal stage. Therefore, we found that the E-NEWS system was able to significantly reduce CPRs and transfers to the ICU on ordinary wards, especially during the intensive period (Figure 2).

False alarms and alarm fatigue are important issues. We sent notifications directly to patients’ in-charge physicians. The E-NEWS system reminded and updated the in-charge physicians, providing the most recent information of clinical deterioration. If we had simply followed conventional rules, in which NEWS ≥ 5 triggers an urgent clinical review and NEWS ≥ 7 triggers a high-level clinical alert, we might have sent many false alarms to physicians [5]. To minimize resistance from physicians, we only selected patients with scores of E-NEWS that had increased to more than 4 in the last 12 h, and with the most recent E-NEWS of ≥ 7. The notification included not only the threshold above 7, but also the deterioration trend of E-NEWS. There were around 22 notifications at 7:00 and 31 at 17:00 every day. If we set the criteria of notifications at only NEWS ≥ 7, there were 86 notifications at 7:00 and 92 at 17:00. The alert system became more precise (Appendix A). It was important that our physicians accepted this alert system. We believe that the mobile text messages sent to physicians contributed to the positive impact that E-NEWS had on patients’ healthcare outcomes.

This study has some limitations. First, we had to arbitrarily replace the missing data using the above-stated measures (i.e., no GCS data or a motor response score of 6). Secondly, we did not collect on-line responses from in-charge physicians who received mobile notifications. Thus, our analysis was restricted to analyze the sensitivity and specificity of the notifications and did not include comments on E-NEWS from the in-charge physicians. Thirdly, we did not include information regarding pharmacological treatment, which might influence the needs of CPR. Lastly, the application of our model into other hospitals or countries was doubtful, due to dissimilar situation and program coding.

## 5. Conclusions

We successfully constructed an E-NEWS system, which automatically provided an hourly status of patients’ clinical deterioration. We integrated this E-NEWS system into real-life clinical practice. Based on our experience of implementing E-NEWS, the early warning system appeared to be an effective platform of communication among healthcare team members. We clearly demonstrated a positive effect of E-NEWS in reducing AEs, especially in CPRs at wards and transfer to ICU from ordinary wards. In the future, further research is needed to explore precise prediction using machine deep learning and big data (clinical information), as well as hourly E-NEWS.

## Figures and Tables

**Figure 1 ijerph-18-04550-f001:**
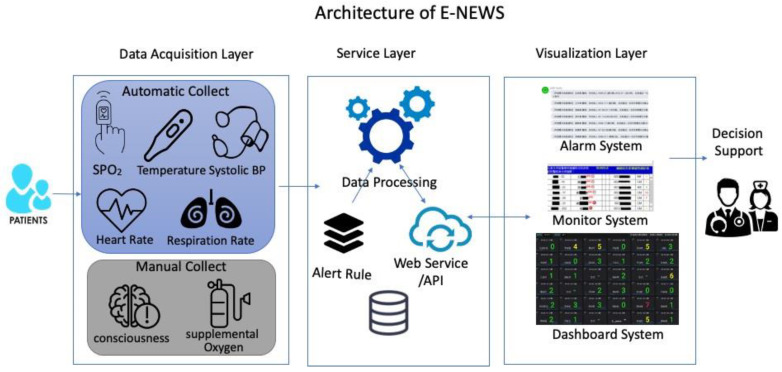
The architecture of E-NEWS comprised three layers: data acquisition, service, and visualization. The data acquisition period involved extracting data that were obtained by nurses. The service layer involved data processing according to the rules of scoring and alerts which we have defined. The visualization layer displayed scores on HIS, NIS, and Dashboard formats.

**Figure 2 ijerph-18-04550-f002:**
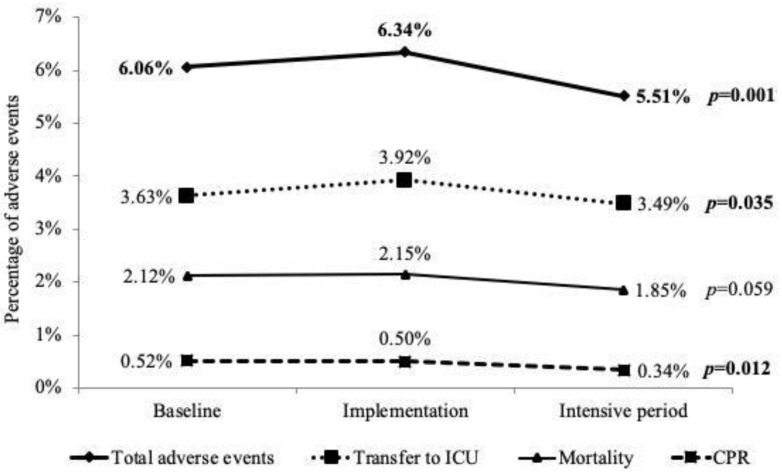
The subtypes of adverse events rates at the 3 stages of implementing online NEWS. Total adverse events decreased after implementation of E-NEWS. The percentages of CPR and transfer to ICU were significantly reduced.

**Table 1 ijerph-18-04550-t001:** Demographic data at the different periods of implementation.

	Total (*n* = 73,674)		Baseline (*n* = 23,543)		Implementation (*n* = 30,035)		Intensive Period (*n* = 20,096)		*p* Value
	*n*	%	*n*	%	*n*	%	*n*	%	
Age (mean ± SD)	57.9	17.1	58.1	17.3	57.8	17.2	57.9	16.8	0.165
Gender-Male	37,543	(50.96%)	12,036	(51.12%)	15,415	(51.32%)	10,092	(50.22%)	0.044 *
Diagnosis groups at discharge									
Cancer	31,462	(42.70%)	9843	(41.81%)	12,945	(43.10%)	8674	(43.16%)	0.003 **
Cardiovascular disorder	12,083	(16.40%)	3874	(16.45%)	4910	(16.35%)	3299	(16.42%)	0.944
Neurological (non-stroke)	11,998	(16.29%)	3788	(16.09%)	4929	(16.41%)	3281	(16.33%)	0.597
Neurological (stroke)	3240	(4.40%)	1040	(4.42%)	1290	(4.29%)	910	(4.53%)	0.452
Respiratory disorder	7791	(10.57%)	2564	(10.89%)	3250	(10.82%)	1977	(9.84%)	<0.001 **
Diabetes	10,012	(13.59%)	3113	(13.22%)	4135	(13.77%)	2764	(13.75%)	0.137
Gastroenterology	9571	(12.99%)	3041	(12.92%)	3895	(12.97%)	2635	(13.11%)	0.823
Renal disorder	6961	(9.45%)	2256	(9.58%)	2839	(9.45%)	1866	(9.29%)	0.572
DNR code	4751	(6.45%)	1515	(6.44%)	1990	(6.63%)	1246	(6.20%)	0.164
Settings of clinical care									0.994
Medical departments	39,452	(53.55%)	12,600	(53.52%)	16,087	(53.56%)	10,765	(53.57%)	
Surgical departments	34,222	(46.45%)	10,943	(46.48%)	13,948	(46.44%)	9331	(46.43%)	

Kruskal–Wallis test. Chi-Square test. * *p* < 0.05, ** *p* < 0.01.

**Table 2 ijerph-18-04550-t002:** Adverse events at different periods of implementation.

	Baseline		Implementation		Intensive Period		*p* Value
	*n*	%	*n*	%	*n*	%	
Adverse events group	1427	(6.06%)	1904	(6.34%)	1108	(5.51%)	0.001 **
Age group							
Age <40	79	(2.12%)	125	(2.62%)	77	(2.58%)	0.294
Age 40–60	429	(5.24%)	567	(5.34%)	334	(4.55%)	0.043 *
Age 60–80	656	(7.24%)	839	(7.31%)	477	(6.14%)	0.003 **
Age ≥80	263	(10.25%)	373	(11.76%)	220	(11.05%)	0.192
Gender group							
Female	507	(4.41%)	665	(4.55%)	389	(3.89%)	0.038 *
Male	920	(7.64%)	1239	(8.04%)	719	(7.12%)	0.027 *
Diagnosis at discharge							
Cancer	656	(6.66%)	874	(6.75%)	522	(6.02%)	0.080
Cardiovascular disorder	566	(14.61%)	780	(15.89%)	446	(13.52%)	0.011 *
Neurological (non-stroke)	425	(11.22%)	585	(11.87%)	345	(10.52%)	0.163
Neurological (stroke)	174	(16.73%)	235	(18.22%)	145	(15.93%)	0.348
Respiratory disorder	406	(15.83%)	602	(18.52%)	339	(17.15%)	0.026 *
Diabetes	361	(11.60%)	496	(12.00%)	300	(10.85%)	0.347
Gastroenterology	317	(10.42%)	349	(8.96%)	218	(8.27%)	0.015 *
Renal disorder	320	(14.18%)	439	(15.46%)	257	(13.77%)	0.219
DNR code	896	(4.07%)	1216	(4.34%)	702	(3.72%)	0.005 **
Settings of clinical care group							
Medical departments	842	(6.68%)	1150	(7.15%)	637	(5.92%)	<0.001 **
Surgical departments	585	(5.35%)	754	(5.41%)	471	(5.05%)	0.464

Chi-Square test. * *p* < 0.05, ** *p* < 0.01.

**Table 3 ijerph-18-04550-t003:** Risk factors of adverse events in univariate and multivariate model analysis.

	Univariate Model		Multivariate Model	
	OR	(95% CI)	OR	(95% CI)
Stage of implementation				
Baseline	ref.			ref.
Implementation	1.05	(0.98–1.13)	1.05	(0.97–1.13)
Intensive period	0.90	(0.83–0.98) *	0.90	(0.83–0.98) *
Age	1.03	(1.02–1.03) **	1.01	(1.01–1.01) **
Gender (Male vs. Female)	1.84	(1.73–1.96) **	1.47	(1.38–1.57) **
Diagnosis group at discharge				
Cancer	1.16	(1.10–1.24) **	2.16	(2.02–2.32) **
Cardiovascular disorder	3.88	(3.64–4.13) **	2.87	(2.65–3.10) **
Neurological (non-stroke)	2.42	(2.26–2.59) **	1.64	(1.39–1.93) **
Neurological (stroke)	3.53	(3.21–3.89) **	2.08	(1.86–2.32) **
Respiratory disorder	4.24	(3.96–4.55) **	2.97	(2.75–3.21) **
Diabetes	2.40	(2.24–2.58) **	0.75	(0.64–0.89) **
Gastroenterology	1.73	(1.60–1.87) **	1.35	(1.24–1.47) **
Renal disorder	3.16	(2.93–3.41) **	1.62	(1.49–1.77) **

Logistic regression with * *p* < 0.05, ** *p* < 0.01.

## Data Availability

Data is contained within the article or Appendix A.

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
