# Peer review of "Implementation of an Electronic National Early Warning System to Decrease Clinical Deterioration in Hospitalized Patients at a Tertiary Medical Center"

_ijerph, 2021, doi:10.3390/ijerph18094550_

Round 1

Reviewer 1 Report

The authors addressed important issue in the domain of healthcare to predict the clinical deterioration of hospitalized patients.

The obtained results were critically evaluated, presented and well discussed. The strong points of this study include 1) datasets were described in sufficient detail to understand in whom the model was intended for use and the sample size seems to be sufficiently large.

However, a major point and some minor points should be addressed:

  • Internal validation for new models is needed. I recommend this reference: Gerry S, Bonnici T, Birks J, Kirtley S, Virdee PS, Watkinson PJ, Collins GS. Early warning scores for detecting deterioration in adult hospital patients: systematic review and critical appraisal of methodology. BMJ. 2020 May 20;369:m1501. doi: 10.1136/bmj.m1501. PMID: 32434791; PMCID: PMC7238890.
  • The authors should consider the TRIPOD (Transparent Reporting of a multivariable prediction model for Individual Prognosis Or Diagnosis) statement, which is a reporting guideline for studies developing or validating prognostic models.
  • It should be also addressed the sample size calculation and the statistical power of this study.
  • The authors should consider use the template from the journal they are submitting the manuscript. In addition, the numeration of all headings and subheadings are not correct.

The findings will be of importance in further clinical practice.

Author Response

Reviewer 1

The authors addressed important issue in the domain of healthcare to predict the clinical deterioration of hospitalized patients.

The obtained results were critically evaluated, presented and well discussed. The strong points of this study include 1) datasets were described in sufficient detail to understand in whom the model was intended for use and the sample size seems to be sufficiently large.

However, a major point and some minor points should be addressed:

Internal validation and external validation for new models is needed. I recommend this reference: Gerry S, Bonnici T, Birks J, Kirtley S, Virdee PS, Watkinson PJ, Collins GS. Early warning scores for detecting deterioration in adult hospital patients: systematic review and critical appraisal of methodology. BMJ. 2020 May 20;369:m1501. doi: 10.1136/bmj.m1501. PMID: 32434791; PMCID: PMC7238890.

The authors should consider the TRIPOD (Transparent Reporting of a multivariable prediction model for Individual Prognosis Or Diagnosis) statement, which is a reporting guideline for studies developing or validating prognostic models.

It should be also addressed the sample size calculation and the statistical power of this study.

Response: Thanks for your suggestion. We have revised our article according to reviewer’s recommendation and added abovementioned article as a reference. (Internal and external validation in page 3, the end of section 2.2. Sample size estimation and statistical power in page 5, the end of section 2.5.)

The authors should consider use the template from the journal they are submitting the manuscript. In addition, the numeration of all headings and subheadings are not correct.

Response: We have modified our article according to the template. Something went wrong during the format transformation in the numeration of all headings and subheadings. We will correct it in revised manuscript.

The findings will be of importance in further clinical practice.

Response: Thanks for your compliment.

Reviewer 2 Report

IJERPH 1171383 peer review. 12th of April, 2021.

Typos : Lines 35, 40, 87, 95, 106, 124, 133, 134, 143, 149, 150. Ok I am going to stop mentioning these typos. Please amen throughout the manuscript.

Introduction is labelled as section 1, but Materials & Methods too… Please amend.
The Section on line 161 is labelled as “1.1” while 1.1 is already given to the section on line 128.

Lines 94-95. Why was consent waived ? This is major concern and may represent ethical breach. Please explain.

Lines 121-125. How can records contain pulse rate over 300 per min ? Similar comments for temperatures below 25 or above 45 °C. This is a major concern regarding records and accuracy of diagnoses.

Line 169. Why is the ‘results’ section labelled as section 1 ? Again, I am going to stop mentioning these.  Please amen throughout the manuscript.

Lines 178-179. Please amend this sentence, as it is difficult to understand.

Line 185. What is an ‘emergent CPR’ ?

Another concern is the lack of records regarding pharmacological interventions in these patients. Indeed, administration of pharmacological treatments in patients should decrease the need of CPR. How was this monitored ? Could it induce a bias in the results presented herein ?

While this E-NEWS systems seems interesting, the lack of details (English translations, program coding, etc…) greatly limit its use in other countries. This is, again, another pitfall.

Author Response

#Reviewer 2

Typos : Lines 35, 40, 87, 95, 106, 124, 133, 134, 143, 149, 150. Ok I am going to stop mentioning these typos. Please amen throughout the manuscript.

Introduction is labelled as section 1, but Materials & Methods too… Please amend.

The Section on line 161 is labelled as “1.1” while 1.1 is already given to the section on line 128.

Response: Something went wrong during the format transformation in the numeration of all headings and subheadings. We will correct it in revised manuscript. Sorry for the mistake.

Lines 94-95. Why was consent waived ? This is major concern and may represent ethical breach. Please explain.

Response: This study was reviewed and approved by the IRB of studied hospital (protocol no./IRB TCVGH No: CE19197B). The data was extracted directly from our hospital’s server. Besides, all patients’ information was anonymized and de-identified prior to analysis. We have explained thoroughly to our IRB and they agreed with waiving of informed consents from participants. The confidential data is well-protected during studied period.

Lines 121-125. How can records contain pulse rate over 300 per min ? Similar comments for temperatures below 25 or above 45 °C. This is a major concern regarding records and accuracy of diagnoses.

Response: Although the values of vital signs were uploaded to the server directly after measurement, it did have missing or extreme values existed in the database. The problem might be due to the error arising during data transfer (machine interface). If we don’t eliminate such records, it will distort the model. Therefore, we have to deal with such values.

Line 169. Why is the ‘results’ section labelled as section 1 ? Again, I am going to stop mentioning these. Please amen throughout the manuscript.

Response: We have corrected all numeration in revised manuscript. Sorry for the mistake.

Lines 178-179. Please amend this sentence, as it is difficult to understand.

Response: Thanks for your suggestion. We will amend this sentence. (In page 5, lower portion of section 3.1.)

Line 185. What is an ‘emergent CPR’ ?

Response: Sorry for the obscure. We have modified related sentence in revised manuscript. (In page 7, line 3 and 6, page 10, conclusion section)

Another concern is the lack of records regarding pharmacological interventions in these patients. Indeed, administration of pharmacological treatments in patients should decrease the need of CPR. How was this monitored ? Could it induce a bias in the results presented herein ?

Response: Thanks for your reminding. We have added more descriptions in the limitation section. (In page 10, the end of discussion section)

While this E-NEWS systems seems interesting, the lack of details (English translations, program coding, etc…) greatly limit its use in other countries. This is, again, another pitfall.

Response: Thanks for your reminding. We’ll also add more descriptions in the limitation section. (In page 10, the end of discussion section)

Round 2

Reviewer 2 Report

IJERPH 1171383 v2. Peer Review. 20th of April 2021.

The authors have made a real effort to answer my concerns.

I still have some doubts about the lack of records for pharmacological treatments adminsitered, or not, to patients. This might represent a bias, but the authors have chosen to include a sentence in the "limitation" section.

Hence, I am confident that this study would be of interest to some readers. Publication of this article should go forward.